# Primary Squamous Cell Carcinoma of the Liver with Good Response to Carboplatin and 5-Flurouracil: A Case Report

**DOI:** 10.3390/medicina58121864

**Published:** 2022-12-17

**Authors:** Hsu-Lin Lee, Chun-Kai Fu, Liang-Yu Chien, Li-Mien Chen

**Affiliations:** 1Division of Hematology and Oncology, Department of Internal Medicine, Taichung Armed Forces General Hospital, Taichung 411, Taiwan; 2Division of Hematology and Oncology, Department of Internal Medicine, Tri-Service General Hospital, National Defense Medical Center, Taipei 114, Taiwan; 3Division of Gastroenterology, Department of Internal Medicine, Taichung Armed Forces General Hospital, Taichung 411, Taiwan; 4Division of Gastroenterology, Department of Internal Medicine, Tri-Service General Hospital, National Defense Medical Center, Taipei 114, Taiwan; 5Division of Clinical Pharmacy, Tri-Service General Hospital Songshan Branch, National Defense Medical Center, Taipei 114, Taiwan

**Keywords:** primary squamous cell carcinoma of the liver, carboplatin, 5-flurouracil

## Abstract

Primary squamous cell carcinoma (SCC) of the liver is a rare disease that is difficult to diagnose until the pathology is confirmed. The age of the patients generally ranges from 18 to 83 years. The pathogenesis of primary SCC of the liver remains unclear and therapeutic guidelines have not yet been established. The overall survival rate may be less than 1 year. The prognosis for patients without surgery is worse than that for patients who undergo surgery. Herein, we report a case of primary SCC of the liver that responded well to intravenous carboplatin and 5-flurouracil (5-FU) with the aim of providing an alternative therapeutic option. A 61-year-old woman with no history of alcohol use disorder, cirrhosis, exposure to hepatotoxic chemicals, or a remarkable family history presented to our hospital with a complaint of epigastric pain, poor appetite, and fatigue, which had occurred 3 days before presentation. Blood tests revealed levels of alpha-fetoprotein of <2.0 ng/mL, carcinoembryonic antigen of 4.39 ng/mL, carbohydrate antigen (CA) 19-9 of 1306.15 U/mL, CA 125 of 66.3 U/mL, CA 153 of 19.7 U/mL, and SCC antigen of 8.5 ng/mL. Computed tomography scans of the abdomen showed a 5.8-cm lobulated soft-tissue mass with central necrosis in segment 6 of the liver, which caused compression of the common hepatic duct. Pathological examination of the masses revealed squamous cell carcinoma with focal glandular differentiation. The patient underwent palliative chemotherapy with intravenous carboplatin 150 mg (day 1) and 5-FU 1000 mg (days 1–4) instead of surgery. After two cycles of chemotherapy, jaundice and liver function improved. The patient was discharged in stable condition and was followed up in our outpatient department. Although the patient refused to undergo surgery, no tumor recurrence or distant metastasis was found during the 8-month follow-up period. This report highlights that neoadjuvant chemotherapy with carboplatin and 5-FU can be considered for primary SCC of the liver before a liver resection.

## 1. Introduction

Primary squamous cell carcinoma (SCC) of the liver is a rare disease that is difficult to diagnose until the pathology is confirmed. According to a reported case series, the ages of the patients range widely from 18 to 83 years, and the male-to-female ratio is approximately 19:16 [1]. The pathogenesis of primary SCC of the liver remains unclear and is generally believed to be associated with hepatic cysts and bile duct stones because chronic inflammation may result in neoplastic transformation [1]. Therapeutic strategies vary from systemic chemotherapy, radiotherapy, and transarterial chemoembolization (TACE) to surgical hepatic resection. However, the prognosis is poor because the tumor is usually diagnosed late. The overall survival rate may be less than 1 year [2]. To our knowledge, the prognosis of patients undergoing surgery is better than that of others; however, systemic chemotherapy may also result in favorable outcomes for those who cannot undergo surgery. Herein, we report a case of primary SCC of the liver that responded well to intravenous carboplatin and 5-flurouracil (5-FU) with the aim of recommending neoadjuvant chemotherapy with carboplatin and 5-FU before a liver resection.

## 2. Case Presentation

The patient was a 61-year-old woman with no history of alcohol use disorder, cirrhosis, exposure to hepatotoxic chemicals, or any remarkable family history. Three days before presentation, the patient experienced epigastric pain, poor appetite, and fatigue. The patient was admitted to our hospital in 2022. Physical examination revealed an icteric sclera and tenderness over the right upper quadrant of the abdomen. Blood tests revealed a white blood cell count of 16,200 cells/mm^3^ with 81.6% neutrophils, hemoglobin level of 10.3 g/dL, platelet count of 403,000 cells/mm^3^, aspartate aminotransferase level of 221 U/L, alanine aminotransferase level of 261 U/L, total bilirubin level of 2.3 mg/dL, alkaline phosphatase level of 423 U/L, gamma-glutamyltransferase level of 355 U/L, C-reactive protein level of 11.53 mg/dL, alpha-fetoprotein level of <2.0 ng/mL, carcinoembryonic antigen level of 4.39 ng/mL, carbohydrate antigen (CA) 19-9 level of 1306.15 U/mL, CA 125 level of 66.3 U/mL, CA 153 level of 19.7 U/mL, and SCC antigen level of 8.5 ng/mL. Other blood tests for hepatitis B and C viruses were negative. Computed tomography (CT) scans of the abdomen showed a 5.8-cm lobulated soft-tissue mass with central necrosis in segment 6 of the liver, which caused compression of the common hepatic duct (Figure 1a). Distention of the gallbladder due to extrinsic compression of the cystic duct was also noted. The patient underwent percutaneous transhepatic cholangiography and drainage, and endoscopic retrograde cholangiopancreatography with implantation of a plastic stent in the left posterior lobe. Intravenous piperacillin/tazobactam (4 g/0.5 g every 8 h) was administered. After her condition stabilized and the infection was under control, we performed ultrasound-guided biopsy of the heterogeneous mass in the right lobe of the liver for pathologic confirmation. Moreover, positron-emission tomography-CT(PET-CT) was used to exclude the presence of other primitive unknown tumors, which concluded mass-forming cholangiocarcinoma (Figure 2). However, hematoxylin and eosin (HE) staining of the liver biopsy revealed squamous cell carcinoma with focal glandular differentiation. Immunohistochemical staining was positive for cytokeratin (CK) 7, p40, and p63. (Figure 3) However, the stain for CK20 and thyroid transcription factor 1 were negative. To identify primary squamous carcinoma lesions, we also performed nasopharyngoscopy, esophagogastroduodenoscopy, colonoscopy, vaginal ultrasound, and whole-body bone scan, all of which showed no significant findings. Based on these findings, we concluded that the tumor was a primary SCC of the liver.

In terms of treatment, we first consulted a general surgeon. However, surgery was deemed to have a high risk of mortality. We decided the patient would not benefit from the surgical intervention due to the poor general condition (Eastern Cooperative Oncology Group (ECOG) 3). Thus, we decided to use palliative chemotherapy with intravenous carboplatin 150 mg (day 1) and 5-FU 1000 mg (days 1–4) instead of surgery. After two cycles of chemotherapy, the jaundice and liver function improved with each passing day. We considered surgery for the patient again, but the patient refused to undergo surgery due to personal reasons. Therefore, the patient was discharged in a stable condition and was followed up in our outpatient department. The course of chemotherapy with carboplatin 150 mg (day 1) and 5-FU 1000 mg (days 1 to 4) was administered every 4 weeks in a cyclic manner. Following this treatment, CT scan revealed a reduction in tumor size compared with that in the previous scan. (Figure 1b) No tumor recurrence or distant metastases were found during the 8-month follow-up period.

## 3. Discussion

Primary SCC of the liver is a rare and sporadically reported malignant disease. There are various clinical symptoms, including abdominal pain, jaundice, weight loss, decreased appetite, and dysphagia; however, these are non-specific [3,4].

The etiology of primary SCC of the liver remains unknown but chronic inflammation, associated with hepatolithiasis or pre-existing cysts has been proposed as the major etiological factor that causes squamous metaplasia, dysplasia, and carcinogenesis of epithelial cells from the biliary tract or cyst wall [2,5,6,7,8]. However, the above hypothesis lacks evidence due to its rare incidence, and further research is needed to determine the true mechanism.

Diagnosis of primary SCC of the liver is based on clinicopathological findings and needs to include features from patient history, clinical examination, and radiological and histopathological investigations. However, diagnosis is difficult because of the lack of specific serum markers. The data regarding serum SCC antigen are not reliable for diagnosis [3]. CT imaging of the liver can provide useful information about the lesion numbers, location, extent, degree of invasion, and whether surgical resection is possible [9]. It is still difficult to discriminate between malignant and benign lesions and infectious disease. In most cases, the diagnosis must be confirmed pathologically.

HE staining of the liver SCC biopsies revealed that the tumor cells were nest-shaped and exhibited abnormal nuclear morphology and keratinization as keratin pearls. Immunohistochemistry revealed that the tumor cells were positive for p63 and p40. Cytokeratin 7 (CK7) is a marker for the bile duct epithelium. In our case study, HE staining was positive for p63 and p40, which was consistent with SCC. In addition, the CK7 stain was also positive, suggesting an association with the carcinogenesis of epithelial cells from the biliary tract. Further studies are required to determine the causes of carcinogenesis.

To date, no therapeutic guidelines have been established for this condition. Systemic chemotherapy, radiotherapy, TACE, and surgical resection have been used to treat the primary liver SCC. Zhang et al. reported that overall survival time was longer in patients undergoing radical surgery than in those undergoing palliative treatment, with median survival times of 17 versus 5 months, respectively [1]. Complete surgical excision of the tumor is strongly recommended. Early surgical resection before tumor invasion of the surrounding liver parenchyma may result in a good prognosis [10]. Boscolo et al. reported a good outcome with neoadjuvant chemotherapy using cisplatin and 5-FU for tumors that are initially unresectable [11]. Although the outcome for patients without the opportunity for surgery is worse, systemic or intra-hepato-arterial chemotherapy is still an alternative option. Limuro et al. reported that intra-hepato-arterial chemotherapy with 5-FU and cisplatin led to a favorable therapeutic effect [12]. Xiao et al. reported a long-surviving patient who had been treated with TACE and adjuvant chemotherapy (gemcitabine and 5-FU) [3]. In addition to traditional chemotherapeutics, immune checkpoint inhibitors (PD-1 inhibitors) have emerged as potential therapeutic agents in recent decades. Wang et al. reported a good response in a patient with primary liver SCC treated with PD-1 inhibitors after surgery [13]. However, there is no strong evidence to support this treatment and further studies are required to prove its efficacy. In our case study, the patient tolerated chemotherapy well and no significant complications were observed. The patient’s jaundice improved and the performance status was upgraded from 3 to 2 on the ECOG scale. The patient is still alive and was followed up for at least 8 months after diagnosis. Compared with the low median survival time in patients who did not undergo surgery, our case shows encouraging clinical results. Although the patient refused to undergo surgery due to personal reasons, neoadjuvant chemotherapy with carboplatin and 5-FU can be considered for primary liver SCC therapy before a liver resection.

## 4. Conclusions

Herein, we report a rare case of a primary SCC of the liver. This case study highlights that SCC should always be considered in patents with hepatic tumors and elevated serum SCC antigen levels. In addition, neoadjuvant chemotherapy with carboplatin and 5-FU treatment are recommended.

## Figures and Tables

**Figure 1 medicina-58-01864-f001:**
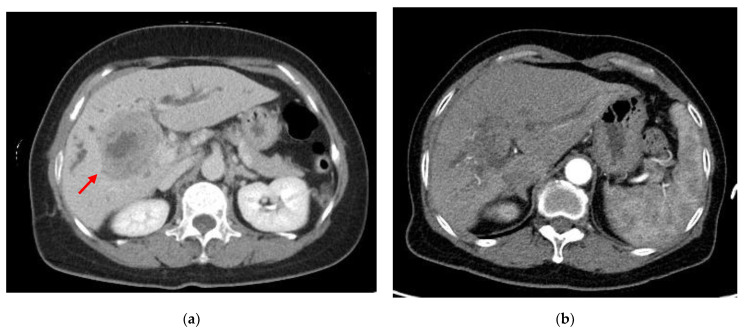
Abdominal computed tomography with contrast medium. (**a**) Scan shows a 5.8-cm lobulated soft-tissue mass with central necrosis in segment 6 of the liver (arrow); (**b**) scan after chemotherapy.

**Figure 2 medicina-58-01864-f002:**
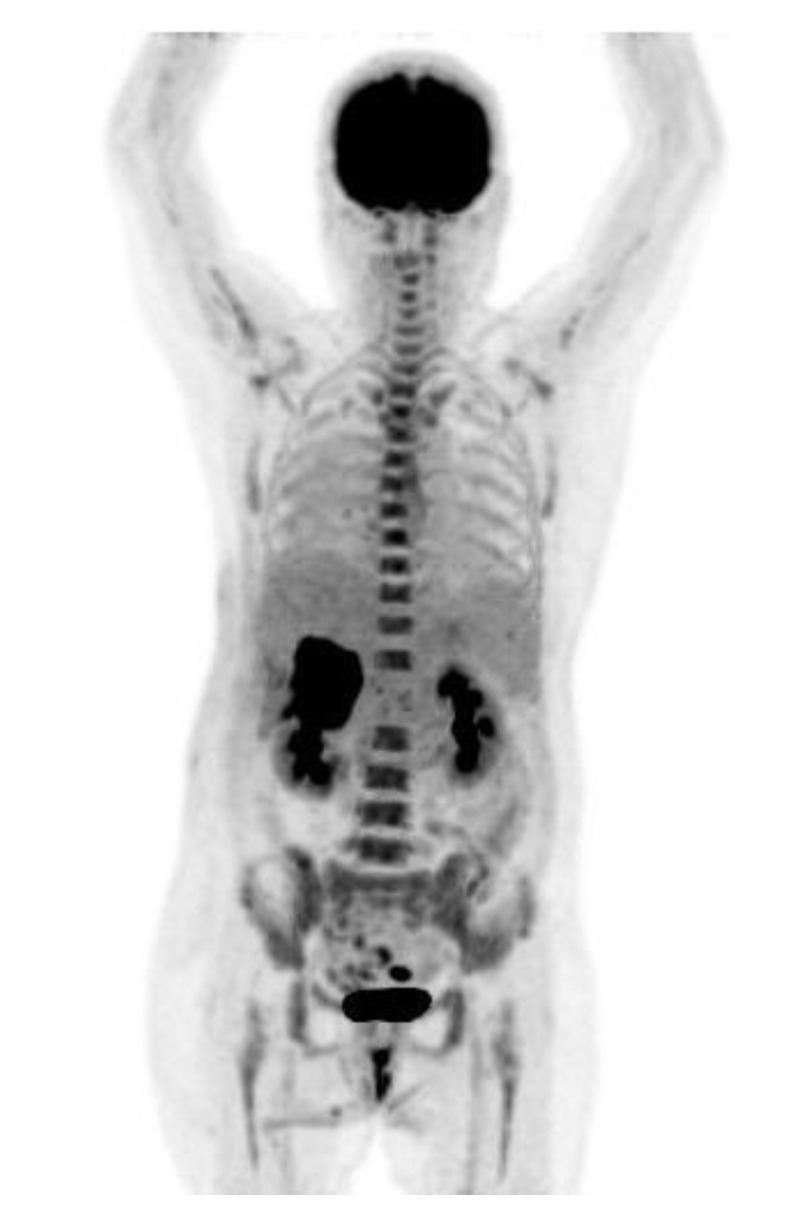
Positron emission tomography scan revealed a mass with intense fluorodeoxyglucose uptake in the right hepatic lobe.

**Figure 3 medicina-58-01864-f003:**
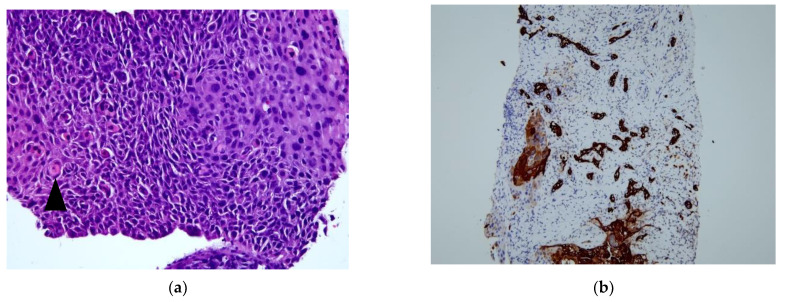
Microscopic evaluation of the lesion. (**a**) Squamous cell carcinoma (SCC) and keratin pearls (arrowhead) observed using hematoxylin and eosin staining (magnification, 200×). (**b**) Immunohistochemistry (IHC) results for cytokeratin 7 showing positivity for tumor cells (magnification, 100×). (**c**) IHC staining for p40 showing positivity for SCC (magnification, 100×). (**d**) IHC staining for p63 showing positivity for SCC (magnification, 100×).

## Data Availability

The data that support the findings of this study are available from the corresponding author, L.-M.C., upon reasonable request.

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
