# Peer review of "Primary Squamous Cell Carcinoma of the Liver with Good Response to Carboplatin and 5-Flurouracil: A Case Report"

_medicina, 2022, doi:10.3390/medicina58121864_

Round 1

Reviewer 1 Report

Dear Authors

thank you for providing this interesting case.

I have a few questions and do only partly agree with your conclusion

You state that radical surgery possibly provided better survival

IN the case report ECOG is reported as 3, maybe due to cholangitis?

In this patient you intervened with Chemo - I do not understand why the patient should be fit for this type of chemo but not surgery? Please elaborate this decision

After chemo you find good control and better ECOG with PR of the disease. Why was surgery not considered then?

The decision for chemo was probably a good choice and your approach using this kind of chemo sensible. But why was surgery in a case with a solitary lesion and PR under Chemo never considered on top? if the patient survives this type of chemo she will be fit for surgery as well.

I would expect to clarify this dilemma in your paper. 

Congratulations on presenting this case, it needs a little more work.

Also I would recommend to follow the patient for a longer time before final submission

Reviewer 2 Report

In this manuscript, Lee. et al. presented a patient with primary squamous cell carcinoma (SSC) of the liver, and the authors also provided detailed information and compelling evidences to support the diagnosis and the efficacy of the palliative chemotherapy. The case with SSC, as the authors say, is a rare disease and has a reference value for clinicians. However, in the paper, some issues need to be addressed.

1.     In Fig.3, figure legend (a) mentioned “arrowhead”, but I cannot see it in Fig.3 (a);

2.     In “Informed Consent Statement” section, there is one patient in the paper, why are there “all subjects”?

3.     There are some language issues, such as “was admitted to in 2022”, “complicationa”, “The patient still alive and been followed up”, et al.

Round 2

Reviewer 1 Report

I think the report has improved and has been clarified in its message.

Still - it needs English language revision.

After that I am fine with publication
